# Fldgen v1.0: An Emulator with Internal Variability and Space-Time Correlation for Earth System Models

Robert Link[1], Abigail Snyder[1], Cary Lynch[2], Corinne Hartin[1], Ben Kravitz[3,4], and Ben Bond-Lamberty[1]

[1]Pacific Northwest National Laboratory, Joint Global Change Research Institute, 5825 University Research Ct., College Park, MD, USA

[2]Connecticut Department of Energy and Environmental Protection, 10 Franklin Square New Britain, CT, USA

[3]Department of Earth and Atmospheric Sciences, Indiana University, 1001 E. 10th St., Bloomington, IN, USA

[4]Atmospheric Sciences and Global Change Division, Pacific Northwest National Laboratory, 902 Battelle Boulevard, Richland, WA, USA

**Correspondence:** Robert Link (robert.link@pnnl.gov)

**Abstract.** Earth System Models (ESMs) are the gold standard for producing future projections of climate change, but running them is difficult and costly, and thus researchers are generally limited to a small selection of scenarios. This paper presents a technique for detailed emulation of Earth System Model (ESM) temperature output, based on constructing a deterministic model for the mean response to global temperature. The residuals between the mean response and the ESM output temperature
fields are used to construct variability fields that are added to the mean response to produce the final product. The method produces grid-level output with spatially and temporally coherent variability. Output fields include random components, so the system may be run as many times as necessary to produce large ensembles of fields for applications that require them. We describe the method, show example outputs, and present statistical verification that it reproduces the ESM properties it is intended to capture. This method, available as an open-source R package, should be useful in the study of climate variability
and its contribution to unertainties in the interactions between human and earth systems.

## 1  Introduction

There are a variety of scientific applications that use data from future climate scenarios as input. Examples include crop and agricultural productivity models (Rosenzweig et al., 2014; Elliott et al., 2014; Nelson et al., 2014), water and hydrology
models (Cui et al., 2018; Voisin et al., 2017), energy models (Turner et al., 2017), and global human systems models (Akhtar et al., 2013; Calvin and Bond-Lamberty, 2018). Earth System Models (ESMs) are the gold standard for producing these future projections of climate change; however, running ESMs is difficult and costly. As a result, most users of ESM data are forced to rely on public libraries of ESM runs produced in model intercomparison projects, such as the CMIP5 (Coupled Model Intercomparison Project) archive (Taylor et al., 2012). Although a few experiments have produced larger ensembles of runs
(e.g. Kay et al., 2015), typically users are limited to a small selection of scenarios with only a handful of runs for each scenario.

This limited selection of scenarios may be inadequate for many types of studies. Users might need customized scenarios following some specific future climate pathway not covered by the scenario library, or they might need many realizations of one or more future climate scenarios.

Examples of research areas for which archival runs might be insufficient include uncertainty studies, in which the multiple realizations are used to compute a statistical distribution of outcomes in the downstream model (Murphy et al., 2004; Falloon et al., 2014; Sanderson et al., 2015; Bodman and Jones, 2016; Rasmussen et al., 2016). Studying tail risk (*i.e.*, the effects of climate variables assuming values in the tails of their distribution, which by definition occurs infrequently in any single scenario run) is another example (Greenough et al., 2001), and studying sensitivity to climate variability is a third (Kay et al., 2015).

In these situations, researchers typically turn to *emulators* to get access to a sufficient quantity of data without having to do an infeasible amount of computation. Climate model emulators attempt to approximate the output a climate model *would have* produced had it been run for a specified scenario. Perhaps the best known emulator algorithm is *pattern scaling*, which develops in each grid cell a linear relationship between global mean temperature $T_g$ and the climate variable or variables being modeled (Mitchell et al., 1999; Mitchell, 2003; Tebaldi and Arblaster, 2014). A variety of enhancements to this basic procedure have been proposed, mostly centering around adding additional predictor variables (*i.e.*, besides just $T_g$) (MacMartin and Kravitz, 2016), adding nonlinear terms to the emulator function (Neelin et al., 2010), or separating the climate state into components, each with its own dependence on the predictor variables (Holden and Edwards, 2010).

Most of these methods are deterministic functions of their inputs, and thus their outputs can be viewed as expectation values for the ESM output. Real ESM output, however, would have some distribution around these mean response values. We will refer to these departures from the mean response generically as "variability." Many of the applications described above are sensitive to climate variability. For example, Ray et al. (2015) found that "Globally, climate variability accounts for roughly a third ($\sim$32--39%) of the observed yield variability" in agricultural crops. Therefore, capturing this variability in emulators is crucial to understanding the behavior of and uncertainties in these applications.

There have been some attempts to add variability to emulators, but producing realistic variability is difficult, due to the complicated correlation structure exhibited by climate model output over both space and time. Typically methods deal with this difficulty by either placing *a priori* limits on the form of the correlation function (Castruccio and Stein, 2013), or by using bootstrap resampling of existing ESM output (Osborn et al., 2015; Alexeeff et al., 2016).

In this paper we describe a computationally-efficient method for producing climate scenario realizations with realistic variability. The realizations are constructed so as to have the same variance and time-space correlation structure as the ESM data used to train the system. The variability produced by the method includes random components, so the system may be run many times with different random number seeds to produce an ensemble of independent realizations. The results in this study are limited to temperature output at annual resolution. Future papers will extend the method to additional output variables, such as precipitation, and to subannual time resolution.

## 2 Method

### 2.1 Notation

In the text that follows, we use underlined bold symbols (*e.g.* $\underline{\mathbf{R}}$) to refer to matrices. Ordinary bold symbols are used for vectors (*e.g.* $\boldsymbol{x}$). When it is necessary to distinguish between column and row vectors, the latter will be marked as the transpose of a column vector (*e.g.* $\boldsymbol{x}^\top$). These vectors represent collections of scalar quantities that bear some relationship to each other in time or space. Because of this, the same variable can appear in both vector and scalar variants, with the vector decoration (or lack thereof) indicating which is meant. For example, $T_g$ is the global mean temperature, a scalar, while $\boldsymbol{T_g}$ is a vector representing a sequence of global mean temperatures.

Occasionally we will add a matrix and a vector; *e.g.*, $\underline{\mathbf{B}} = \underline{\mathbf{A}} + \boldsymbol{x}$. This should be interpreted to mean that the vector $\boldsymbol{x}$ is to be added to each row of the matrix $\underline{\mathbf{A}}$. Therefore, the length of $\boldsymbol{x}$ must be equal to the number of columns in $\underline{\mathbf{A}}$. This *broadcast* convention is slightly nonstandard mathematically, but it is common in programming languages that support matrix arithmetic (*e.g.* the *numpy* package for python), and simplifies certain expressions that will come up in the derivation.

### 2.2 Input

Our method requires a collection of ESM model output to train on. Any model can be used, and by switching out the input data the method can be tuned to produce results representative of any desired ESM. For all of the results in this paper we have used the CESM(CAM5) (Community Earth System Model (Community Atmosphere Model)) output from the CMIP5 archive (Taylor et al., 2012). We used surface temperature data from all available 21st century runs for all four Representative Concentration Pathway (RCP) emissions scenarios (RCP2.6, RCP4.5, RCP6.0, and RCP8.5), for a total of 9 runs, each 95 years in length. These data were averaged to annual resolution, for a total of 855 global temperature states.

To keep clear the distinction between the data produced by the emulator and the ESM data used to train the emulator, we will refer to the ESM data as "synthetic measurements" (when referring to the data as a whole) or "cases" (when referring to individual frames in the data), while the terms "results" and "model output" will be reserved for the data produced by the emulator.

Throughout the discussion, we will treat each temperature state as a vector, with each grid cell providing one entry in the vector. The ordering of the grid cells within the vector is arbitrary, but consistent throughout the entire calculation. The entire set of synthetic measurements will be grouped into the input matrix $\underline{\mathbf{O}}$, with the cases in rows and grid cells in columns. In the input data used for this study, each case is 288 (longitude) $\times$ 192 (latitude), for a total of 55296 grid cells. Therefore, in this case, $\underline{\mathbf{O}}$ has dimension $855 \times 55296$.

We will also derive from the input an operator for computing the area-weighted mean of a grid state. We denote this vector by

$$\boldsymbol{\lambda} = \frac{1}{S}\sin(\boldsymbol{\theta}), \tag{1}$$

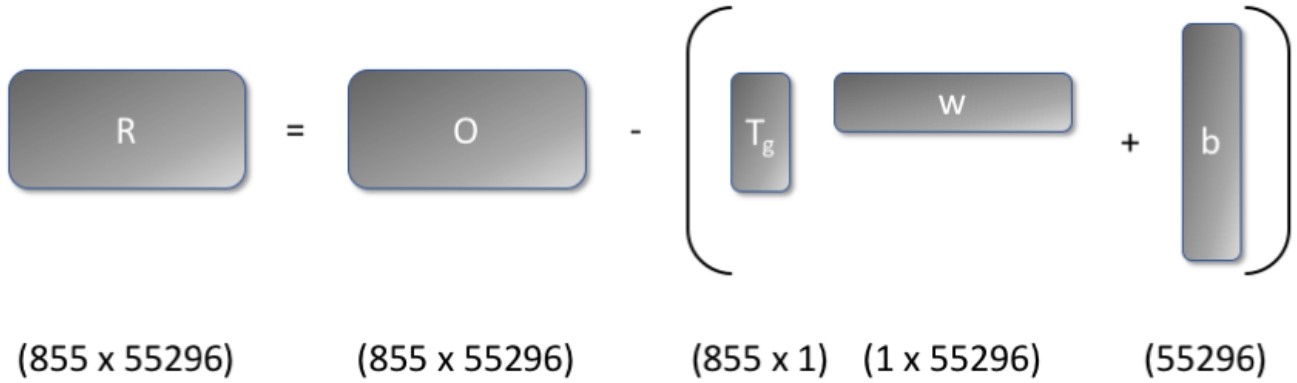

**Figure 1.** Schematic of the residual calculation showing the shapes of the matrices involved. The result of the outer product $T_g w^\top$ is an $855 \times 55296$ matrix. The vector $b$ is added to this matrix using the broadcast convention described in section 2.1

where $\theta$ is the polar angle (*i.e.*, *colatitude*) of each grid cell, and $S$ is the sum of all the area weights across the entire grid. When defined this way, the global mean temperature for a grid state $x$ is $T_g = \lambda^\top x = x^\top \lambda$. Similarly, the matrix-vector multiplication $T_g = \underline{O}\lambda$ produces a vector of global mean temperature values for the entire input data set.

### 2.3 Mean response model

Our basic procedure will be to construct a deterministic model for the mean response to global temperature. The residuals between the mean response and the synthetic temperature fields will be taken as representative of the variability in the ESM and used to construct variability fields that will be added to the mean response to produce the final product.

In principle the mean response could be calculated using any of the emulation techniques described in section 1. For illustrative purposes we will stick with a simple linear pattern scaling using a linear regression variant similar to that described in Mitchell et al. (1999). Using standard least-square regression techniques we compute vectors of weights $w$ and biases $b$ (each of these vectors has length equal to the number of grid cells) such that the mean response field $m$ for global mean temperature $T_g$ is given by

$$m(T_g) = T_g w + b. \tag{2}$$

This formula can be applied to the entire input data set, with $T_g w$ becoming the outer product $T_g w^\top$ to produce the residual matrix

$$\underline{R} = \underline{O} - \left( T_g w^\top + b \right), \tag{3}$$

which will be used to construct the variability model. This calculation is shown schematically in Figure 1. Conversely, the variability fields generated will be added to the mean response (*i.e.*, the last term of equation (3)) to generate absolute temperature fields.

## 2.4 Generating variability

The matrix of residuals, $\underline{\mathbf{R}}$, characterizes the variability in the input data. We deem a generated variability data set to be realistic if it matches the distribution of residual values in each grid cell and the space and time correlation properties of the residuals. Our task, therefore, is to generate a random field with specified distribution and correlation properties.

To capture the time correlation we will make use of the Wiener-Khinchin Theorem (Champeney, 1973, § 5.4). This theorem states that given a function $g(t)$ and its Fourier transform $G(f)$,

$$\mathcal{F}(C(g)) = |G(f)|^2, \tag{4}$$

where $C(g)$ is the time autocorrelation function of $g(t)$, and $\mathcal{F}(C)$ is the Fourier transform of $C$. The salient feature of equation (4) is that the right-hand side of the equation depends only on the magnitudes of the elements of $G$, not their phases (recall that the results of a Fourier transform are complex numbers with both magnitude and phase). Therefore, we can generate an alternate function $g'$ by setting $|G'| = |G|$, selecting the phases of $G'$ at random, and taking the inverse Fourier transform. When $g'$ is constructed this way, the Wiener-Khinchin Theorem guarantees that $g$ and $g'$ will have the same autocorrelation function.

In theory we could use a similar technique to capture the spatial correlation; however, in practice the spherical geometry of the spatial domain makes this difficult. Moreover, it is not just the spatial correlation properties that matter, but also the locations at which spatially correlated phenomena occur. Therefore, we capture spatial correlations by using principal components analysis (PCA) to express the grid state as a linear combination of basis vectors that diagonalize the covariance matrix of the system.

$$\boldsymbol{x}(t) = \sum_{i=1}^{L} \phi_i(t)\hat{\boldsymbol{x}}_i, \tag{5}$$

where

$$\left. \begin{aligned} \hat{\boldsymbol{x}}_i^\top \hat{\boldsymbol{x}}_j &= 0, \\ \mathrm{cov}(\phi_i, \phi_j) &= 0, \end{aligned} \right\} \quad \text{if } i \neq j. \tag{6}$$

The $\hat{\boldsymbol{x}}_i$ are called *empirical orthogonal functions* (EOFs) (Kutzbach, 1967) and are computed using singular value decomposition (SVD) (Golub and Van Loan, 1996, § 2.5.3). The $\phi_i(t)$ are the *projection coefficients* for the grid state vectors. The second property in equation (6) is of particular interest for this application. Because the covariances of the projection coefficients for different EOFs are zero, we can choose them independently. In particular, when applying the phase randomization procedure described above, we can apply it to each $\phi_i$ independently because all of the spatial correlation properties of the system have been absorbed into the definition of the EOFs.

In practice, it is convenient to force all of the basis vectors except for one to have area-weighted global means of zero, so that all of the variability in the global mean is carried by a single component. This property is useful because it allows us to control how much the generated variability distorts the global properties of the mean response field it is being added to. To

accomplish this, we introduce a small modification to the EOF decomposition procedure. We define the zeroth basis vector $\hat{\boldsymbol{x}}_0$ to be the global mean operator, normalized to unit magnitude:

$$\hat{\boldsymbol{x}}_0 = \frac{\boldsymbol{\lambda}}{\sqrt{\boldsymbol{\lambda}^\top \boldsymbol{\lambda}}}. \tag{7}$$

We force $\hat{\boldsymbol{x}}_0$ to be a basis vector by subtracting from each residual vector its projection onto $\hat{\boldsymbol{x}}_0$ and performing the SVD on the
modified residuals. This procedure forces all of the basis vectors to be orthogonal to $\hat{\boldsymbol{x}}_0$. Since this vector is proportional to the global mean operator $\boldsymbol{\lambda}$, this orthogonality property guarantees that all of the other basis vectors will have zero global mean. Therefore, if $\phi_0(t) = 0$, then the global means of the mean response fields will be unaffected when the generated residual fields are added. On the other hand, if it is desirable to change the global means, perhaps because they were generated by a simple climate model (Hartin et al., 2015; Meinshausen et al., 2011) that produces smoother results than real ESMs, then that can be
done by setting $\phi_0$ appropriately.

     The typical use of PCA in many fields, including climate modeling, is for dimensionality reduction. In such applications the next step after computing the EOFs would be to identify and keep a small set of EOFs that capture the majority of the variability and to throw away the rest. In this case, dimensionality reduction is *not* our goal. Rather, we have used the EOF decomposition only to separate the residual field into components that are uncorrelated over time. Therefore, we keep the full
set of EOFs and their projection coefficients. The sole exception is for components for which the singular values produced by the SVD procedure are very small. There are generally 1 or 2 such components, and keeping them can cause problems with roundoff error, so these are dropped.

     At this point we are ready to apply the Wiener-Khinchin Theorem. We compute the discrete Fourier transform (DFT) of the $\phi$ from equation (5): $\Phi(f) = F(\phi(t))$. We then compute $\Phi^\star(f)$ such that $|\Phi^\star| = |\Phi|$, but we choose the phases of $\Phi^\star$ to be
uniform random deviates on the interval $[0, 2\pi]$. From this we can reconstruct $\phi^\star(t)$ as the inverse DFT of $\Phi^\star(f)$. Finally, we construct the variability field using equation (5), replacing $\phi$ with $\phi^\star$.

     The steps in the variability generation algorithm are summarized in Table 1.

## 3    Results, Analysis, and Validation

### 3.1    Model output and performance

To illustrate the algorithm, we have produced four independent variability fields by applying the algorithm to the input data described in section 2.2. Training the emulator (*i.e.*, read-in and analysis of the ESM input) took approximately 143 seconds on a midrange workstation. Each temperature field took 3–4 seconds to generate.

     Figure 2 shows a single time slice for each of the variability fields (i.e., the temperature field, with the mean response field subtracted out). The time series these slices were taken from could be used as an ensemble to study the effects of variability on
the downstream models that are consumers of these sorts of climate projections.

     The spatial structure in the variability is readily apparent. Temperature perturbations occur on scales of roughly 40–60 degrees of arc. Some features, such as the one seen in the low-latitude eastern Pacific, appear in all of the frames, with greater

**Table 1.** Summary of steps in the variability generation algorithm described in section 2

1. Select and fit the mean response model.

2. Construct residual field $\underline{\mathbf{R}}$ by subtracting mean response from ESM output (equation (3)).

3. Orthogonalize residuals with respect to EOF-0 (equation (7)).

4. Perform the EOF analysis on the residual field.

5. Compute the DFT $\Phi$ of the residual field's projection coefficients onto the EOF basis.

6. Compute a new Fourier transform $\Phi^\star$ such that $|\Phi| = |\Phi^\star|$ and the phases of $\Phi^\star$ are chosen randomly, uniformly on the interval $[0, 2\pi)$.

7. Compute the projection coefficients $\phi^\star$ of the variability field as the inverse DFT of $\Phi^\star$.

8. Compute the variability field as $\boldsymbol{x}(t) = \sum_{i=0}^{N} \phi_i^\star(t) \hat{\boldsymbol{x}}_i$.

or lesser strength, or, in one case, with opposite sign. Other features, such as the cool patch over northern Europe in the third frame, have no apparent analog in the other realizations.

We can get a sense of the behavior of the variability fields over time by looking at the power spectral density of the EOFs (fig. 3). Two trends are immediately apparent. First, the total power present in each EOF decreases dramatically after the first few EOFs (fig. 3). The first 10 components together account for 49% of the total power, and the first 50 components account for 72%. Notwithstanding this observation, the long tail of EOFs makes a nontrivial contribution to the result. The last 400 EOFs collectively make up a little over 1% of the total power, and as we shall see below, all of the small-scale variability is contained in these components.

The second observation is that the power spectrum whitens (becomes more uniform across frequencies) considerably (Fig. 4), such that only a few of the most prominent EOFs have any significant periodic signature. One interpretation of this observation is that there are only a few consistently repeatable periodic phenomena represented in the surface temperature data of this ESM. The rest of the variability, although highly structured spatially, does not have a lot of temporal structure. The components with significant periodicity account for roughly a third of the total variability signal. In other words, although periodic oscillations are a prominent component of the variability, most of the variability appears to be of the uncorrelated, interannual sort.

In Figure 5 we show the power spectral density for the first nine EOFs. EOF-1 has power primarily at long periods, indicating a pattern of variability that is largely locked in at the beginning of a run, but which varies from one run to the next. EOFs 2, 3, and 5 show evidence of periodicity on time scales ranging from 3 to 5 years.

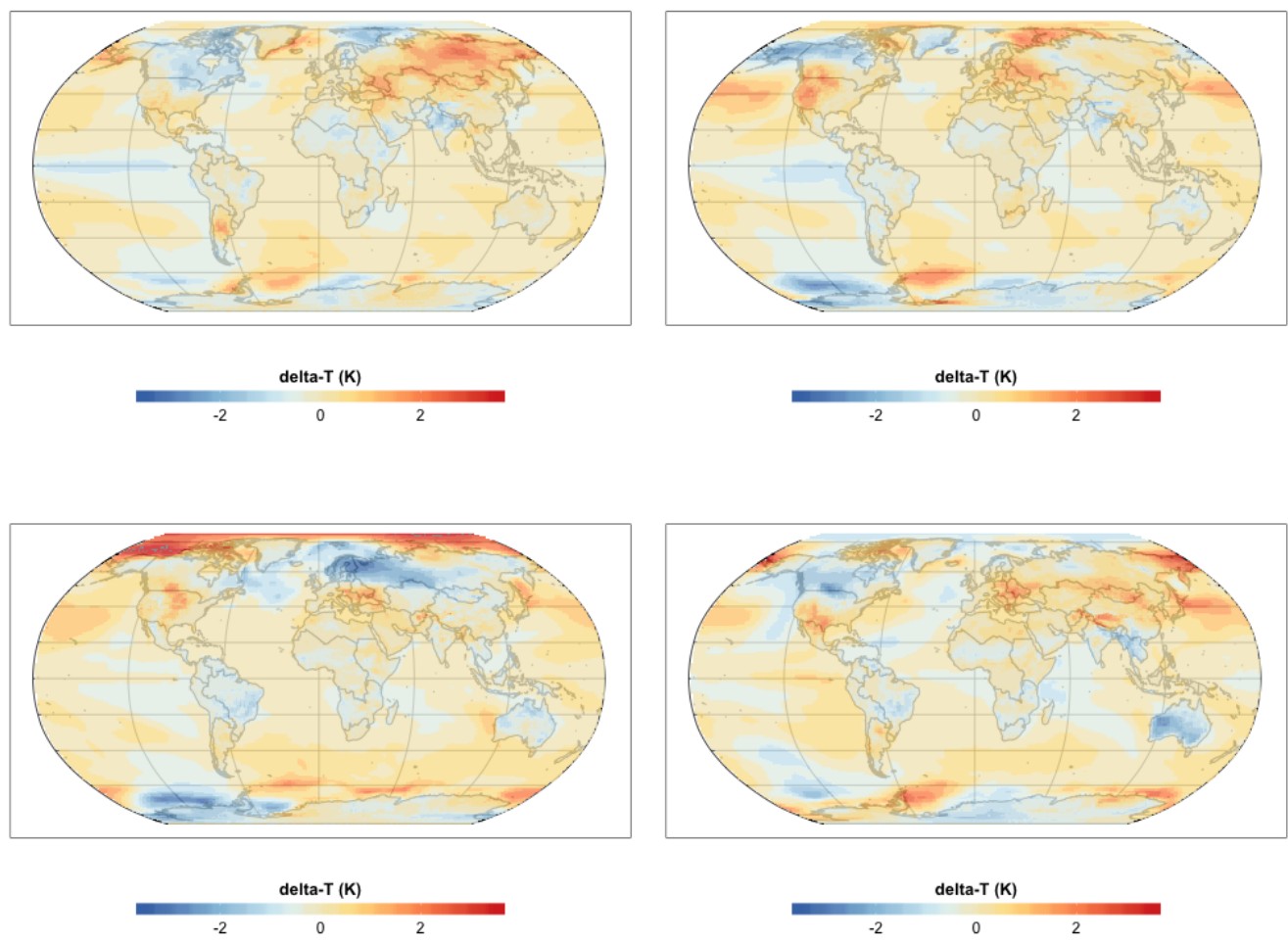

**Figure 2.** Year 2025 snapshot for variability fields generated using the procedure described in section 2.4. Each field is a different randomly generated realization of the temperature field's departure from the mean response (sec. 2.3). The sequences these frames were drawn from could be used as an ensemble of future climate scenarios for studying sensitivities or uncertainties in models that use climate data as inputs.

Figure 6 visualizes the spatial patterns represented by the first 6 EOFs, and Figure 7 visualizes some of the lower power EOFs. These plots show that the scale of the features gets progressively smaller as the power decreases. For example, in EOF-3 there is a complex of positive and negative associations that spans nearly the entire Pacific Ocean. The features visible in EOF-25 are roughly continental scale, while the features in EOF-50 are about half that size. By EOF-400 the feature size is in the hundreds of kilometers, and the lowest power EOF, EOF-853, shows variations a few grid cells in size.

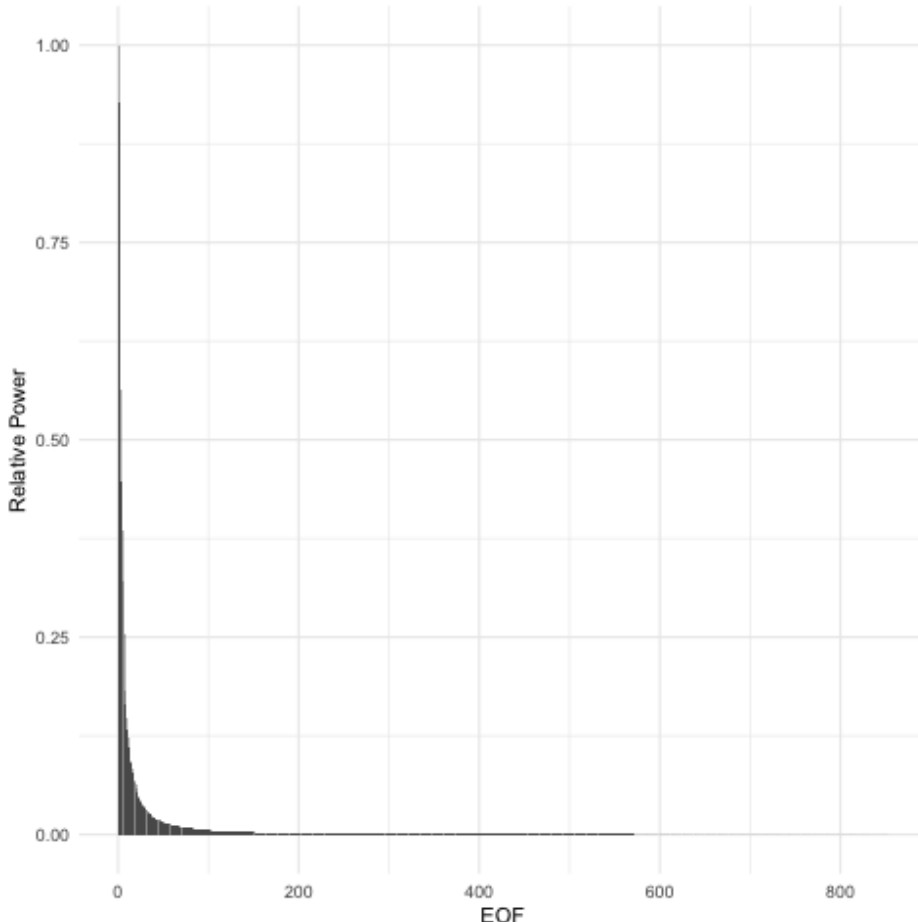

**Figure 3.** Relative power for each EOF. Roughly half of the total power is contained in the first 10 EOFs. The aggregate power for all EOFs beyond 400 is 1% of the total.

## 3.2   Statistical equivalence to ESM input

The time series produced by this method are designed to match three key statistical properties of the ESM data used to train the emulator:

1. Distribution of values in a grid cell over time and between realizations.

2. Correlation between values in different grid cells.

3. Time autocorrelation of spatially correlated patterns of grid cells.

In this section we perform a series of statistical tests to verify properties 1 and 2. Property 3 is guaranteed by the Wiener-Khinchin Theorem, and so we do not test it statistically.

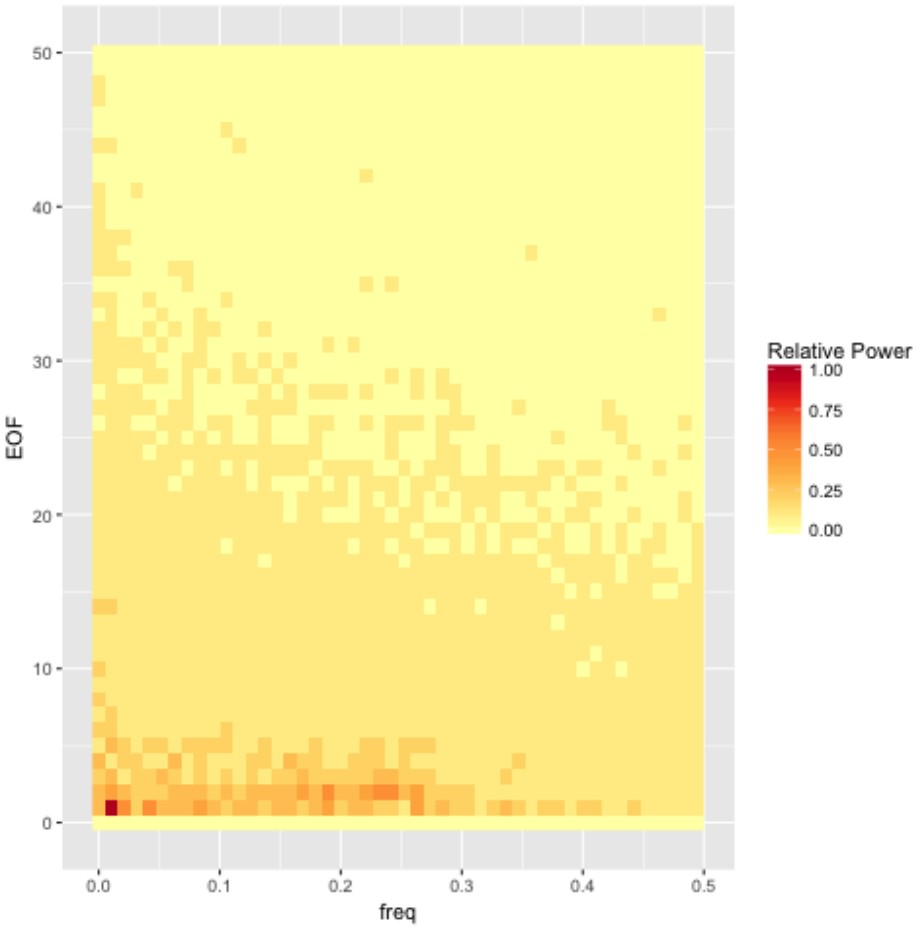

**Figure 4.** Heat map of power spectral density (PSD) for the first 50 EOFs. The trend of decreasing total power and more uniform spectral density continues for the remaining EOFs beyond EOF-50.

### 3.2.1 Statistical tests of variability field properties

The generation procedure described in this paper does not strictly guarantee that the generated fields have the desired statistical properties; therefore, we turn to statistical tests of some of the key properties. Testing for the *absence* of an effect is tricky. One cannot simply run a hypothesis test and, seeing a lack of a positive result, conclude that there is no effect. The procedure we have adopted is to focus on tests that can be run in each grid cell (or, in one case, for each pairwise combination of EOFs). We can consider two competing hypotheses:

**H1** The statistic being tested is the same in the generated data as in the input data.

**H2** The statistic being tested differs in the generated data by some *de minimis* value from the input data.

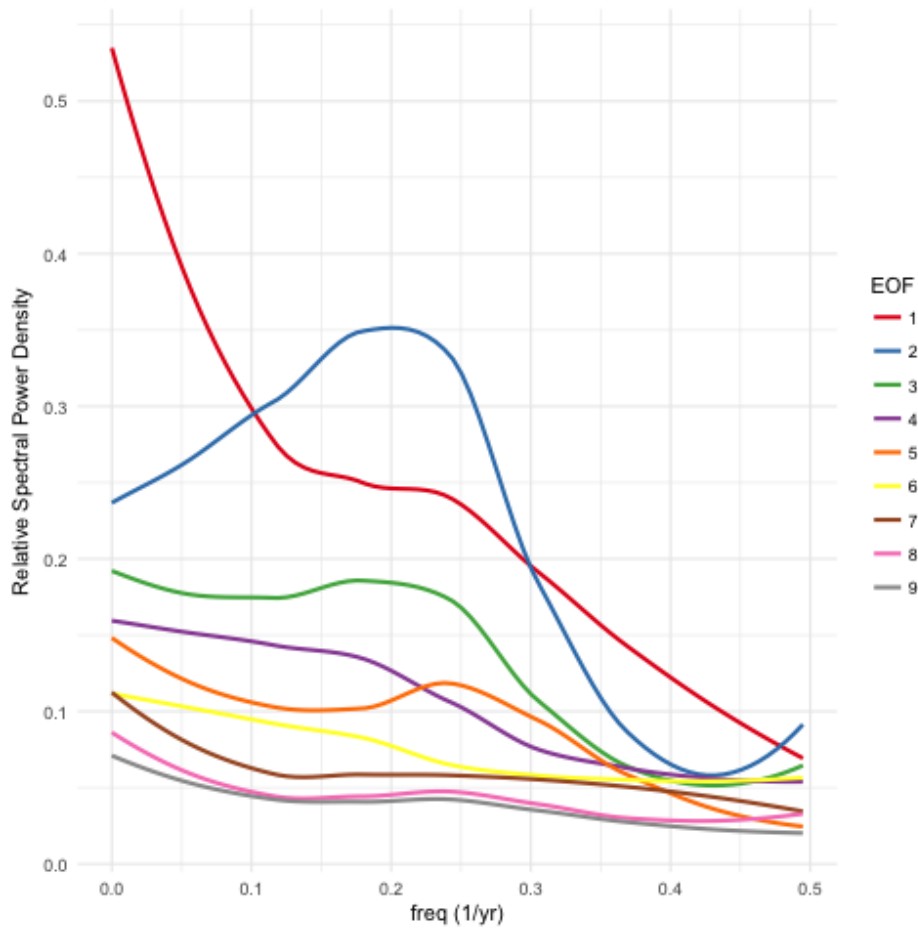

**Figure 5.** Smoothed power spectral density (PSD) for the first 9 EOF basis functions. EOFs 2, 3, and 5 show peaks in the PSD, indicating quasiperiodic behavior on 3–5 year time scales. EOF-1 has most of its power at low frequencies, indicating that this component is approximately (though not exactly) constant over the course of a single ESM run.

The expected numbers of positive results under these hypotheses are just the p-value (H1) and the power (H2) of the test, each multiplied by the number of tests performed. By observing which of the two hypotheses the actual number of positive results agrees with more closely, we can decide which of the two hypotheses is more likely. The philosophy underlying this procedure is that although we cannot prove that there is *no* statistical difference between the generated and input data, if we can show that an upper bound on the effect size is small enough to be ignorable in practice, then that is sufficient.

All of the statistical tests described in this section were performed on an ensemble of 20 generated fields, each with 95 one-year time steps, for a total of 1900 model outputs in the tests that operated directly on the generated data. For the test that

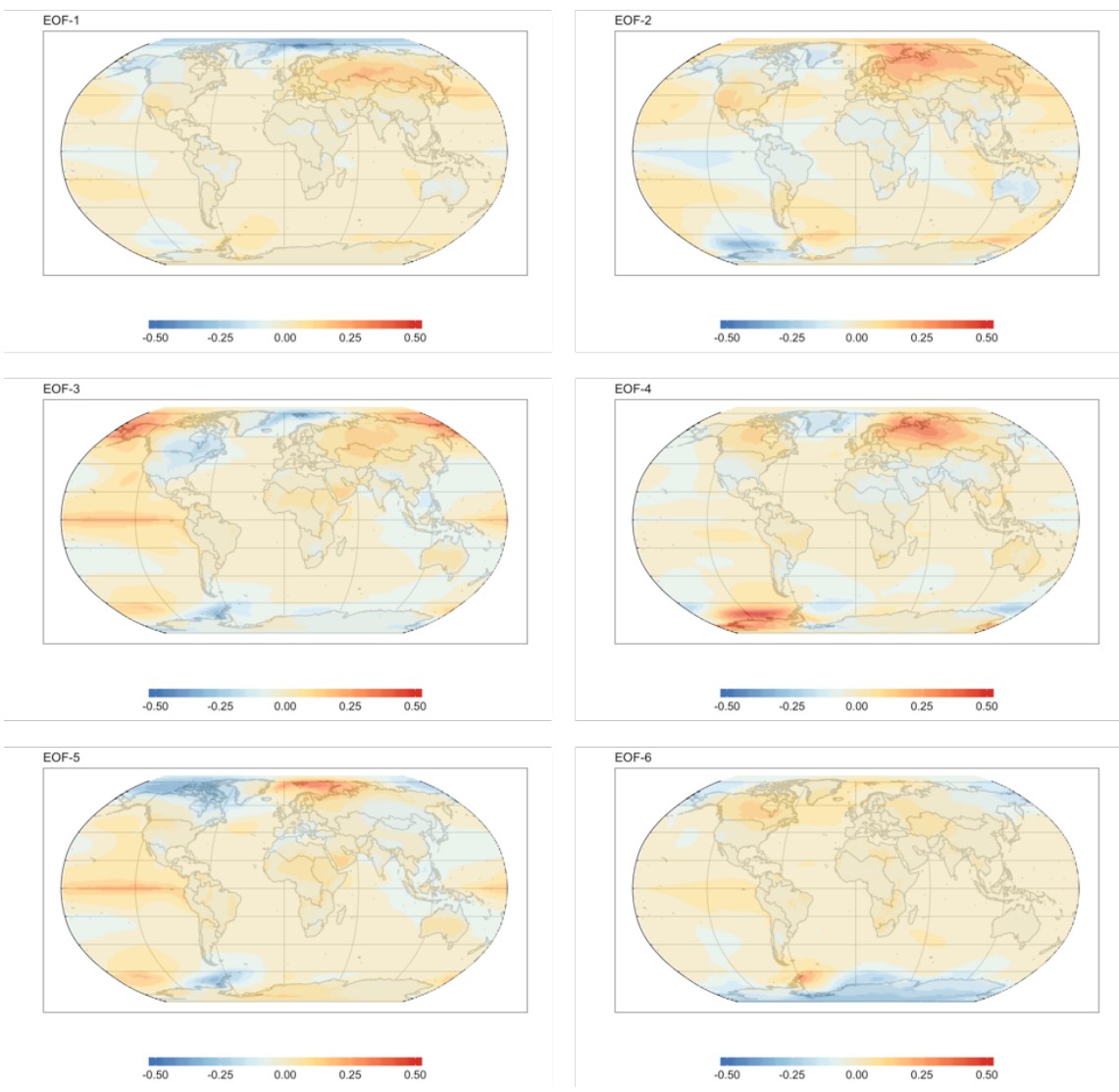

**Figure 6.** Spatial visualizations of the EOF1-6 basis functions. EOF grid cell values are scaled such that the magnitude of the largest value is 1. These components capture large-scale patterns of variability. EOFs 2, 3, and 5 all feature a temperature anomaly in the eastern Pacific. These same components can be seen in figure 5 to have some periodicity on 3–5 year time scales, suggesting that they may be rooted in physical processes in the ESM the model was trained on.

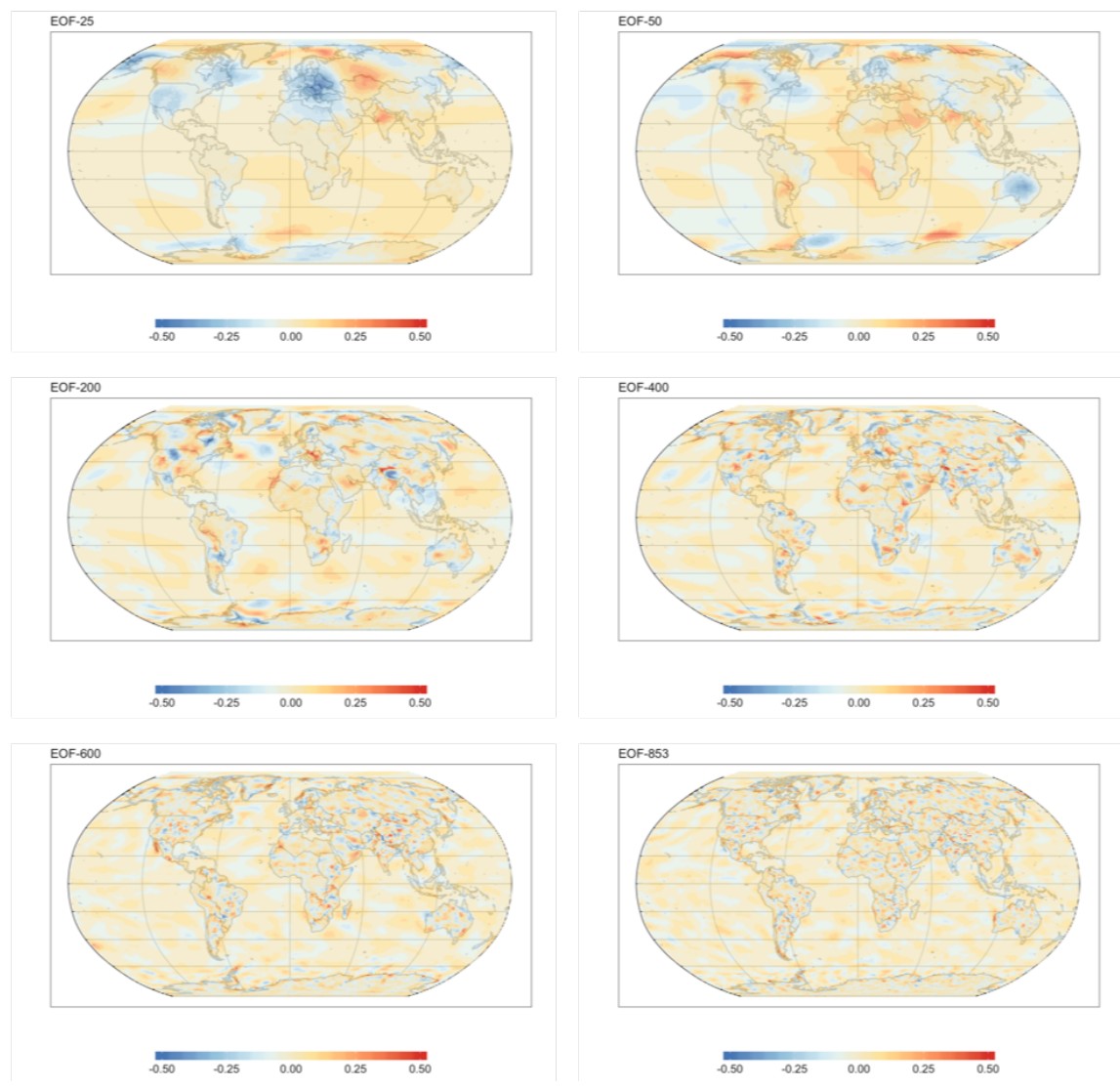

**Figure 7.** Spatial visualizations of higher EOF basis functions. EOF grid cell values are scaled such that the magnitude of the largest value is 1. The characteristic scale of temperature fluctuations decreases for functions later in the series. Thus, EOFs 25 and 50 show features at about half the scale of those shown in figure 6, while features in EOFs 200 and 400 are roughly one quarter the scale. By the time we get to the last few hundred EOFs, features are just a few grid cells in size, resulting in patterns that might be thought of as spatially structured noise.

**Table 2.** F-test power for several hypothetical percentage differences between input and output variance.

| Variance Difference | F-test Power |
|---|---|
| 1% | 0.05 |
| 2.5% | 0.07 |
| 5% | 0.13 |
| 10% | 0.37 |

**Table 3.** Pearson test power for several hypothetical correlation coefficients between $\phi$ for different EOFs.

| Correlation Coefficient | Pearson Test Power |
|---|---|
| 0.01 | 0.07 |
| 0.05 | 0.59 |
| 0.10 | 0.99 |

operates on the $\phi$ values, each temperature grid time series had to be tested separately, for a total of 95 samples per test. In each case the threshold p-value used for the tests was $0.05$.

The first property we will examine is the variance of the distribution of grid cells. We used the F-test of equality of variances to perform this test. In order to be valid, the F-test requires the samples being tested to be normally distributed. We test for this property separately below. Table 2 gives the power (*i.e.*, expected fraction of positive results) for several hypothetical percentage differences in variance between the ESM and generated fields. The actual fraction of positive results was approximately $2 \times 10^{-4}$, which is much smaller than the p-value of $0.05$.

It may seem surprising that the fraction of positive results was so much smaller than the number expected from the p-value of the tests. This result can be explained by observing that the derivation of the p-value assumes a particular model for H1. Specifically it assumes that the generated data and the reference data (*i.e.*, the ESM input) come from *populations* with exactly equal variance. We cannot observe population variances directly; instead we observe the variances of samples from those populations. The variances of such samples can vary quite a bit from the variance of the underlying population, and so we expect to see some fairly large differences between the variances of input grid cells and the corresponding variances of output grid cells. The F-Distribution tells us just how large we might reasonably expect those discrepancies to be.

Our model results, on the other hand, are *not* being generated by sampling from a population. Instead, they are generated by a process that seeks to replicate the variances of the reference data exactly. If it were completely successful at doing so, then all of the variances would be identical to their counterparts in the reference set, and there would be precisely zero positive results. In actuality, there are some slight discrepancies, but these are much smaller than the ones assumed in the formulation of H1. Therefore, we see many fewer positive results than would be expected based on the p-value used in the tests.

Our second test concerns the covariance between grid cells. Testing for equal, nonzero covariances directly is challenging, but we can transform the results into a form that is more readily testable. Starting from equation (5) we can show that for two

grid cells $x_m$ and $x_n$

$$\text{cov}(x_m, x_n) = \sum_i \text{var}(\phi_i)\hat{x}_{im}\hat{x}_{in} + \sum_{i \neq j} \text{cov}(\phi_i, \phi_j)\hat{x}_{im}\hat{x}_{jn}, \tag{8}$$

where $\hat{x}_{im}$ is the $m$th component of $\hat{\boldsymbol{x}}_i$. The corresponding expression for the generated data is the same, except that the $\phi$ are replaced by $\phi^\star$. For the input ESM data, the construction of the EOFs guarantees that $\text{cov}(\phi_i, \phi_j) = 0$, when averaged over the
input data. Thus, the grid cell covariances of the generated data will match those of the ESM data if, averaged over runs of the generator:

$$\text{var}(\phi_i^\star) = \text{var}(\phi_i) \qquad \qquad \text{for all } i, \text{ and} \tag{9}$$
$$\text{cov}(\phi_i^\star, \phi_j^\star) = 0 \qquad \qquad \text{for all } i \neq j. \tag{10}$$

The first of these two conditions is guaranteed by the generation procedure. Parseval's Theorem (Champeney, 1973, ap-
pendix E) states that for each of the $\phi_i$ (and likewise for the $\phi_i^\star$),

$$\sum_{t=1}^{N_t} (\phi_i(t))^2 = \sum_{k=1}^{N_t} |\mathcal{F}_k(\phi_i)|^2 . \tag{11}$$

Since our procedure ensures $|\mathcal{F}_k(\phi_i^\star)| = |\mathcal{F}_k(\phi_i)|$, this guarantees that the condition in equation (9) holds.

To test the condition in equation (10) we used Pearson's correlation test. Table 3 gives the power of the test for various correlation coefficients for the alternative hypothesis. The actual fraction of positive tests, over the pairwise combinations of
EOFs, was $0.05$, or roughly what we would expect from the p-value used in the test. From these observations we can conclude that the upper bound on possible correlation coefficients between the $\phi$ is somewhere between $0.01$ and $0.05$.

The final statistical test concerns whether the generated residuals are normally distributed. Apart from being necessary to ensure the validity of the F-tests above, a normal distribution is desirable per se because we expect the temperature residuals to be normally distributed. This test is more challenging to perform than the rest because there is no obvious way to define
an effect size to use in calculating the power. Instead, we must determine a reasonable nonnormal distribution to use as the benchmark for deviations from normality.

To arrive at such a distribution, consider how the generated residual fields are calculated. The value $x$ of the residual temperature in each grid cell is produced by summing over all EOFs and all Fourier components. Since the phases of the Fourier components are chosen randomly, this amounts to a sum over uniform random deviates, which by the Central Limit Theorem
will be asymptotically normally distributed. Any deviations from normality will be due to having insufficient terms in the sum to reach that asymptotic behavior. Such a distribution would appear truncated compared to the normal distribution, since the sum of uniform random deviates has hard minimum and maximum values. The Beta distribution, $B(n_1, n_2)$ also has these properties. When $n_1 = n_2 = n$, the distribution is symmetric and approaches a Normal distribution as $n$ increases. We adopted the $B(5,5)$ distribution, shown in Figure 8, as our representative distribution for a *de minimis* effect size.
We used the Shapiro-Wilk test of normality to evaluate the normality of the grid cell distribution. For this sample size, the power of the test for distinguishing between a $B(5,5)$ and a Normal distribution is $0.998$. The actual fraction of grid cells that

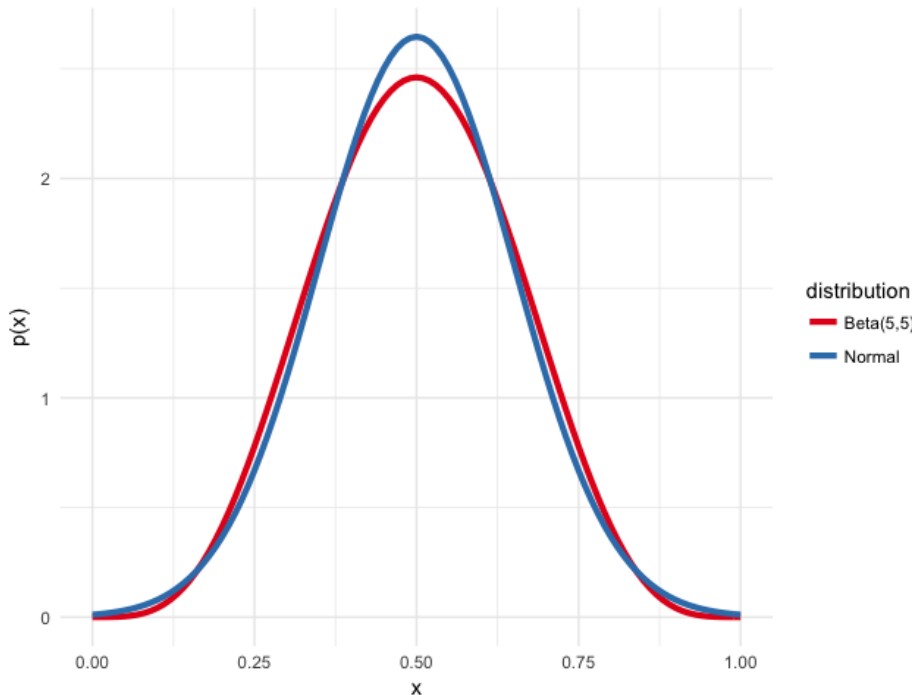

**Figure 8.** Comparison of the Beta(5,5) distribution and a Normal distribution with equal variance. The beta distribution is zero outside of the depicted range, while the normal distribution asymptotically approaches zero. Although the difference between these two distributions is small, the Shapiro-Wilk test can easily distinguish them.

showed a positive result was $0.06$, indicating that if there is any nonnormality, it is almost certainly smaller than the difference between a normal distribution and a $B(5,5)$ distribution.

### 3.2.2 Commentary on statistical properties

Property 3 deserves additional comment because it is explicitly *not* equivalent to matching the time autocorrelation function
5 of individual grid cells. We chose to focus on autocorrelation of spatial patterns rather than on grid cells because the only way to preserve the autocorrelation of grid cells would be to force a constant phase difference between EOFs. This assumption doesn't seem particularly realistic and isn't supported by the input data. Limiting the treatment of time autocorrelation to the EOFs ensures that to the extent that EOFs represent physical phenomena they occur with the right frequency spectra, while not overly constraining the phase relationships between modes.

10 The properties enumerated above ensure that, when using the generated data to drive an ensemble of downstream models and compute statistics on those results, the scale of the fluctuations produced, their spatial location and extent, and their periodic character, if any, will be faithfully reproduced, allowing reliable calculations of variance in outcomes, return times of extremes,

and regional differences in impacts. Therefore, we expect a technique like this to be invaluable for studies of the contribution of variability to uncertainty in climate effects and feedbacks.

Supporting such uncertainty studies was our primary purpose in developing this tool, but the analysis in section 3.1 suggests additional possibilities. A byproduct of the procedure to generate variability fields is that we develop quite a few statistics that could be used to characterize the ESM used to train the emulator. Thus, the training stage of the emulation procedure could also function as a diagnostic package for ESMs. For example, the high power at low frequencies for the first 10–15 EOFs (Fig. 4) was unexpected and might be of interest for further study.

### 3.3 Overfitting the mean response

There is one important pitfall to watch out for when using this method to learn the behavior of an ESM; viz., one must take care not to allow the mean response model to overfit the ESM data. The more complex the model, the greater the danger of overfitting, but even simple models like the linear regression used here can overfit. Consider EOF-1 and its power spectrum, depicted in figure 5. The power spectrum's strong peak at $f = 0$ means that the coefficient $\phi_1$ of the component is nearly constant within a single run of ESM data. Therefore, if we were to train the model on just a single run (*i.e.*, a single realization of a single scenario), this component would be absorbed into the mean response, causing it to be reproduced identically in all generated temperature fields. In fact, this is precisely what happened in early versions of this work, where we trained the emulator on a single ESM run. EOF-1 only began to appear in the variability fields once we expanded the input data to include the full suite of CESM(CAM5) runs from CMIP5.

Therefore, it is essential to include enough independent ESM runs in the training data to ensure that the mean response model will not capture fluctuations that are idiosyncratic to a particular run. Exactly how many runs are needed will depend on the complexity of the mean field response model. For a relatively simple model, such as the linear model used in this paper, as few as three independent runs (i.e., one more than the number of parameters per grid cell) should provide reasonable protection against absorbing variability features into the mean response model. Conversely, mean response models with many parameters per grid cell would require more independent inputs. In case of doubt, cross-validation should be used to diagnose possible overfitting. Along similar lines, the input data should include runs for scenarios that span the entire range of future scenarios that the system will be used to emulate. This practice ensures that the mean response model will not be called upon to extrapolate beyond the range of conditions it was trained on.

### 3.4 Underfitting the mean response

Several readers of early versions of this work questioned the decision to fit the mean response model over the entire range of RCP scenarios, speculating that this practice would result in a mean response model that represented a sort of compromise amongst the various RCPs in the input data, fitting none of them particularly well. If the mean response model were to be underfit in this way, then the residuals from the misfitting would be lumped in with the variability and subjected to the randomization procedure described in section 2.4. It was suggested that the long-period behavior of EOF-1 might be evidence that this was happening.

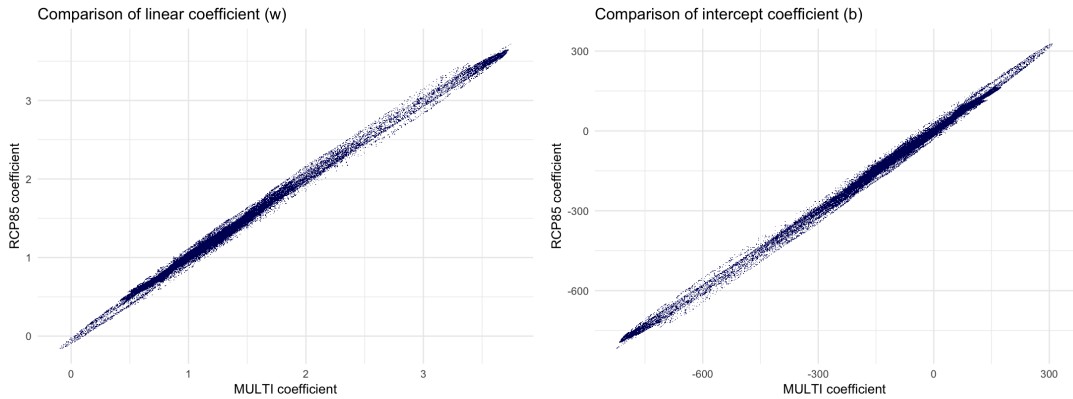

**Figure 9.** Comparison between coefficients of the mean response model for the RCP85 and MULTI emulators. For both the linear term $w$ (left) and the intercept term $b$ (right) the two models are nearly identical.

Throughout the rest of this section we will refer to this collection of hypotheses as the *Compromise Conjecture*, or CC for short. We know that the CC is true to some extent, since it seems unlikely that the relationship between global and local temperatures in these models has *no* dependence on the specifics of the warming scenario. One solution to the CC would be to fit separate emulators for each of the RCP warming scenarios; however, for scenarios that do not correspond exactly to an RCP, we still need to generate fields using an approximate mean response, and we will need to know how much of an error we are making. Therefore, the question we must answer is, are the effects of using a compromise model acceptably small in the context of the other approximations used in the emulator's design?

To investigate this question, we fit two more emulators to subsets of the data. The first of these used only the three ensemble members for the RCP-8.5 scenarios. We designated this emulator "RCP85". The second fit used three ESM runs covering the RCP-2.6, RCP-6.0, and RCP-8.5 scenarios. We designated this emulator "MULTI". Our first test was to compare the mean field models for these two emulators. Figure 9 shows a grid cell by grid cell comparison of the $w$ (linear) and $b$ (intercept) coefficients for the two models, from which it can be see that the two mean response models are very similar.

We can quantify just how similar the two models are by fitting linear models predicting the RCP85 coefficients from the corresponding MULTI coefficients. When we do this, we find that the average ratio between the RCP85 and MULTI $w$ terms is 0.994, with an $R^2$ of 0.999. Most of the residuals are within +/- 0.02 of 0 (for a coefficient that ranges approximately from 0–3). For $b$, the relationship is nearly as good; the coefficient ratio is 0.987, with an $R^2$ of 0.998. Most of the residuals are between -5 and +6 (the scale of this variable is considerably larger than the scale for $w$: approximately $-650 - +300$.)

From this result alone, we see that the mean response models for these two emulators are virtually identical, making it extremely unlikely that CC effects are an appreciable source of error in the MULTI emulator. For this reason, description of additonal tests of CC effects, along with source code and results have been relegated to the data and analysis code archive cited in section 4.

## 3.5 Assumptions

As with most emulation schemes, this one makes certain assumptions about the models it is trying to emulate. The most important assumption is that the ESM outputs can be linearly separated into a temperature-dependent component (what we've been calling the "mean field response") and a time-dependent component (the "variability"). Notably, we assume that the temperature response is independent of the temperature history. This assumption, though common in emulator studies, is dubious. The assumption can be partially negated by including additional predictor variables in the mean field model (*e.g.* Joshi et al., 2013; MacMartin and Kravitz, 2016). At the same time, the second assumption implies that the internal dynamics of the ESM are unaffected by the specifics of the external forcing, which is certainly debatable.

A related assumption is the assumption of stationarity. The variability fields produced by this method have stationary statistical properties. Some research has suggested that the variability is likely to change with increasing global mean temperature (Murray and Ebi, 2012). This sort of phenomenon could be added to our method by introducing a global mean temperature-dependent scale factor. Such a factor would be applied in between steps 7 and 8 in Table 1.

## 4 Conclusions

Having a computationally efficient method for generating realizations of future climate pathways is a key enabler for research into uncertainties in climate impacts. In order to be fit for this purpose, a proposed method must produce data with statistical properties that are similar to those of Earth System Models, which are currently the state of the art in projecting future climate states.

In the preceding sections we have described such a method, and we have shown that it reproduces key statistical properties of the Earth System Model on which it was trained. Specifically, it produces equivalent distributions of residuals to the mean field response and equivalent space and time correlation structure. The method is computationally efficient, requiring under 10 minutes to train on the input data set used for the results presented here. Once training is complete, generating temperature fields takes just a few seconds per field generated.

As a result, we believe the method will be extremely useful for the impacts studies it was designed to support. Currently, the method is limited to producing temperature only, and at annual resolution. However, we believe that the method can be readily extended to other climate variables and to shorter time scales. These extensions will be the subject of follow-up work.

*Code and data availability.* Software implementing this technique is available as an R package released under the GNU General Public License. Full source and installation instructions can be found in the project's GitHub repository (https://github.com/JGCRI/fldgen). Release version 1.0.0 of the package was used for all of the work in this paper.

The data and analysis code for the results presented in this paper are archived at https://doi.org/10.5281/zenodo.1183640.

*Author contributions.* Link designed the algorithm, developed the fldgen package and performed the analysis of the results. Lynch ran early versions of the algorithm and performed analysis on those results. Snyder performed the theoretical analysis of the algorithm's properties, which informed the statistical analysis. Hartin, Kravitz, and Bond-Lamberty advised the project and provided feedback on early drafts of the paper. Link prepared the manuscript with contributions from all coauthors.

5  *Competing interests.* The authors declare that they have no conflict of interest.

*Acknowledgements.* This research is based on work supported by the US Department of Energy, Office of Science, as part of research in the Multi-Sector Dynamics, Earth and Environmental System Modeling Program. The Pacific Northwest National Laboratory is operated for DOE by Battelle Memorial Institute under contract DE-AC05-76RL01830.

This research was supported in part by the Indiana University Environmental Resilience Institute and the Prepared for Environmental
10  Change grand challenge initiative.

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
