# Peer review of "Computationally Efficient Emulators for Earth System Models"

_Geoscientific Model Development, 2018_

## Short Comment (SC1) · 2 May 2018

Dear authors,

in my role as Executive editor of GMD, I would like to bring to your attention our Editorial version 1.1: http://www.geosci-model-dev.net/8/3487/2015/gmd-8-3487-2015.html This highlights some requirements of papers published in GMD, which is also available on the GMD website in the 'Manuscript Types' section: http://www.geoscientific-model-development.net/submission/manuscript_types.html. In particular, please note that for your paper, the following requirement has not been met in the Discussions paper:

- "The main paper must give the model name and version number (or other unique

identifier) in the title."

Please provide the name and a version number of the Emulator in the title of your revised manuscript. Note, that a name and a version number are important to identify your specific developments.

As explained in
https://www.geoscientific-model-development.net/about/manuscript_types.html   GMD is encouraging authors to upload the program code of models (including relevant data sets) as supplement or make the code and data of the exact model version described in the paper accessible through a DOI (digital object identifier). In case your institution does not provide the possibility to make electronic data accessible through a DOI you may consider other providers (eg. zenodo.org of CERN) to create a DOI. Please note that in the code accessibility section you can still point the reader to how to obtain the newest version.

Yours, Astrid Kerkweg

———————————————

---

## Author Comment (AC1) · 23 May 2018

We will make this change when we submit our revised manuscript in response to reviewer comments. The new title will be "Fldgen v1.0: A Computationally Efficient Emulator for Earth System Models". Version 1.0 of the package already has a DOI (10.5281/zenodo.1183347), which we will add to the code and data availability section in the revision.

---

## Referee Comment (RC1) · Anonymous Referee #1 · 5 Jun 2018

General comments I think this idea and motivation of the problem presented in this manuscript is interesting. However, I am very confused about what exactly has been done. The paper is written in a far too technical manner and many parts of the language used leads the reader to be confused. I even found the mathematical and statistical part of the paper confusing. If I as a statistician am finding this confusing there is no way a non-statistician will be able to understand what is going on. At this time, I recommend that the manuscript be rejected in order to give the authors enough time to rewrite it. Specific comments to help with this rewrite are given below.

Major comments

[1] Please don't use 'emulator' at all in your manuscript. What you have implemented is a 'surrogate model' or 'metamodel'. An emulator is a type of surrogate model /

metamodel which is an interpolator and gives a probability distribution for outputs corresponding to inputs it is not trained at. See [1] for details. I understand that others in the earth system modelling community have used 'emulator' in the same way as you, but whoever started using it first and passed down this definition is incorrect for doing so.

[2] Your literature review in the introduction is very limited. This would be fine if the journal was very narrowly focused, but GMD has a very broad appeal. Statisticians from the UQ (Uncertainty Quantification) community have, for example, done a lot research on surrogate modelling methods for very expensive models like an ESM. Names that come to mind are Nathan Urban (Los Alamos), Jonty Rougier (Univ. of Bristol), Michael Goldstein (Durham univ.). A series of workshops were held in Cambridge earlier this year (http://www.newton.ac.uk/event/unq/workshops) which will help you find statisticians working in this field. Non-statisticians like David Sexton at the Met office in the UK are also working on quantifying uncertainty of ESMs. There are probably lots of other research that you can also mention, but above is just a start.

[3] Please use more standard mathematical notation for defining vectors and matrices. In particular, please don't use |x> and <x| for column and row vectors. I have never seen this notation being used before. It is much more standard to use x for a column vector and xT (i.e. the transpose of x) for a row vector.

[4] Please also stick to normal conventions for matrix and vector algebra, e.g. page 2 line 28 you state "Occasionally we will add a matrix and a vector e.g. B = A + |x>. Please do not do this! I know that you explain what you mean when you do this, but mathematically it is not the correct way of doing things because you're effectively defining |x> to be a vector sometimes and a matrix and other times. If you want to add a matrix with a vector in the way you describe then define a new matrix which has as its columns or rows the vector you want. This is a much clearer way of defining things.

[5] Page 3, line 8. You state "We refer to the ESM data as "observations" . . . ". NO, NO,

[Figure]

NO! Please do not do this. I spent about an hour reading your paper thinking you were using real observations and then I realized I had not properly read this very important line in your methods. If you want to use your ESM output as measurements then call them "synthetic measurements". Everyone knows what this is. But if you just state measurements or observations, we think you're using the real thing.

[6] There are lots of subject specific words / jargon used here which you assume that the reader knows the meaning of because you don't define them. Examples include: 'linear pattern scaling', 'discrete Fourier Transform', 'randomizing the phases of G', 'spatial coherence', etc. . .

[7] For things that you define you don't give enough detail. For example, with EOFs (statisticians use principal components but it means the same thing) you need to say that we normally only choose the first n EOFs where n is determined such that most of the variance (or power as you mentioned) is explained. Only when we get to the results section do you talk about the number of EOFs that you're using.

[8] I was also completely lost with section 2.4. You seem This seems to be key part of the methods, so really needs to be explained better. You make statements that make no sense to the non-specialist – e.g. in the lines prior to equation 7 you state that you're making some minor modifications to the standard procedure, but why? And what is a zeroth basis vector and why are you defining it in this way. These things I read and go "okay", but I have no idea why you're doing this. This is just one example of many scattered throughout the manuscript.

[9] There's no motivation or justification for doing what you're doing. At the end of the introduction, you state that there's a need to quantify the uncertainty in ESM output. Fine. But then you jump straight into your approach of quantifying this uncertainty by generating it based on an error covariance matrix that is derived from defining the temporal and spatial correlation of the ESM output. Why is this a good way of defining the ESM uncertainty?

[10] Following on from the previous point, this seems to be a major flaw in this paper. Normally when we do uncertainty analysis, we propagate the uncertainty from the inputs (e.g. uncertainty in model parameters) through to the uncertainty in the ESM outputs. Instead, you seem to be using the spatial and temporal correlations of the ESM output as a means to generate the uncertainty in the ESM output with the meta-model (or "emulator" as you call it) as the vehicle for carrying out the extra ESM runs. Perhaps I'm mistaken and this isn't what you're doing. If I am mistaken then the fact that I have misunderstood this is a major problem. If you want people to be interested in your research, you first need to communicate it clearly and (sometimes) simply to them.

[11] I am confused with how you train your metamodel (or "emulator" as you call it). When I train metamodel, this often involves multiple runs of the computationally expensive model (i.e. the ESM in your case). In your paper, it seems that you just need one run of the ESM to train the metamodel. Is this correct? Again, this isn't how we normally train metamodels so you need to be really clear about this. In fact there should really be a whole section in the methods explaining everything about how you constructed the metamodel. Maybe you feel what you've written is enough, but I'm just generally confused so you need to lay things out much more logically and clearly at the very least.

[12] Table 1 makes little sense to me. If I really concentrate I can probably understand what's going on, but you could help the reader by using less complicated words or phrasing it in a simpler way.

[13] The results section seems to be too short (less than one page). Most papers I read have at least 3 or 4 subsections within the results section. These subsections have their own titles and help navigate the reader through the different aspects of the results. At the moment, the results just seems like a list of things. Your results show flow more like a story. You also don't really give a lot of detail, e.g. just one sentence for figure 5? What's the point of having it in there?
[14] I didn't really read through the discussion in depth, but in section 4.1.1 (and may other subsections) you describe further results that were carried out. Thse should be in the results section. The purpose of the discussion section is: (a) to give an explanation for why your results look the way they do; (b) to put your results in context of other comparable studies. I see a bit of (a) in the discussion but no evidence of (b).

Minor comments

- Panels of figures: label them with letters. E.g. Figure 1a would refer to panel at the top left of figure 1.

- Figures: captions lack enough details.

- When submitting for review, it's more helpful to put all the figures and tables at the end of the manuscript. This makes it easier for the reader to refer to a particular figure when reading a particular part of the results.

- There are probably other minor comments but I think the major changes should be made first.

Reference [1] O'Hagan, A., 2006. Bayesian analysis of computer code outputs: A tutorial. Reliability Engineering & System Safety, 91(10-11), pp.1290-1300.

---

## Referee Comment (RC2) · Anonymous Referee #2 · 13 Aug 2018

**1   General comments**

This paper describes an earth system model "emulator" for quickly simulating multiple realistations of earth system model output. This model will be useful for studies in which large ensembles are necessary to evaluate effects, or for probabilistic evaluations such as how likely a given emissions pathway results in a global mean temperature exceeding some pre-defined threshold. The spatial nature of the model also allows investigations into regional effects. As suggested by the authors I can see an application where this model could be applied to the output of simple climate models to introduce temporal and spatial variabilty.

In general the paper is well-written and reasonably easy to follow. The software itself is

publicly available and the examples given in the paper are reproducible. However, the language used is quite mathematical for a GMD paper. I think this could be addressed without loss of quality or conciseness.

Also, as suggested by the first reviewer, this is not an emulator in the strict sense. See Lee et al 2011 https://www.atmos-chem-phys.net/11/12253/2011/acp-11-12253-2011.pdf, fig. 2 for a graphical representation of an emulator in the 1-dimensional case.

I also agree with the first reviewer in that, ESM outputs are not "observations". "ESM outputs" would suffice. A related point is that the model simulates global mean surface temperature from GCMs (general circulation models/global climate models - choose your favourite acronym) rather than ESMs. The CMIP5 definition of an ESM includes an interactive carbon cycle, going from emissions to concentrations to forcing to temperature. GCMs skip the emissions step, running from prescribed concentrations that have been calculated from a simple model, e.g. MAGICC, as they were in CMIP5.

**2   Specific comments**

In the introduction, the application of the model to extreme events is given as a justification for its creation. However, the model only produces annual mean temperature output in each grid cell. I am not aware of an extreme indicator that uses annual mean temperatures. Such indicators are usually calculated from daily climate model output (see Zhang et al 2011, 10.1002/wcc.147). This would be a natural extension to this model, but in its current form it is not capable of analysing "extremes" in the usual sense.

Section 2.3: I don't disagree with the authors about the notation convention: I understand the broadcasting concept used in their convention and agree it aids readability. I do find it hard to follow the equations though. If we have

$$|T_g >= \mathbf{O}|\lambda >$$

then this suggests to me that $|T_g >$ is a column vector of shape 855 x 1 formed by multiplication of $\mathbf{O}$ (855 x 55296) by $|\lambda >$ (55296 x 1). In eq(2) you have $T_g|w > +|b >$. Is $T_g$ (not bracketed in eq(2)) times $|w >$ a column vector times a column vector? How is this defined? And then in equation 3, there is $|T_g >$ (a column) times $w$ (a row), which I think is 855 x 855, then added to $|b >$ (855 x 1)? and subtracted from $\mathbf{O}$ (55296 x 855 - but how is this broadcasted?) If there are no typos in these equations, it would be helpful here to put in a diagram of the matrix dimensions in the equations 1 to 3.

$\sigma$ values in table 1 and p5 line 9. I think these are the singular values of $\mathbf{R}$, but it is not really explained what these are or what they mean. This paragraph could do with some expansion of the key terms (rank deficient, discrete Fourier transform). Does dropping EOFs where $\sigma < \sigma_{threshold}$ guarantee full rank?

Section 3: Can the four images in figure 1 be interpreted as ensemble members? If so, it would be good to state this.

figures 4-6 and associated discussion in lines 24-28 on page 6: The periodic variability in EOFs 2, 3 and 5 - could these have a physical interpretation? For example there seems to be an El Nino style feature in EOFs 3 and 5. On the other hand, is there any evidence that the lower EOFs are not just noise?

Section 4.2 got me thinking that as the model is trained on the RCP outputs, is there any difference in the results when taking just the set of realisations from RCP2.6 and RCP8.5? Certaintly across ESMs, the variance across models increases with increasing global mean temperature. It would therefore not be correct to use a variability model that is trained on RCP8.5 for low forcing scenarios or those with a peak and decline. I note the authors address this in section 4.3, but I wonder if they have tested this.

**3 Technical corrections**

page 5, line 3: allow > allows

page 6, line 3: 143 seconds. What is the machine architecture here?

---

## Author Comment (AC2) · 7 Sep 2018

**1  Major comments**

1.    Please don't use 'emulator' at all in your manuscript. What you have implemented is a 'surrogate model' or 'metamodel'. An emulator is a type of surrogate model / metamodel which is an interpolator and gives a probability distribution for outputs corresponding to inputs it is not trained at. See [1] for details. I understand that others in the earth system modelling community have used 'emulator' in the same way as you, but whoever started using it first and passed down this definition is incorrect for doing so.

[Figure]

The reviewer's point is well taken; however, as the reviewer points out, the term "emulator" is firmly entrenched in the earth system modeling community. Moreover, it is the term used for this kind of model in other papers in this journal. Therefore, we have elected to stick with the terminology that is customary among our target audience.

2.       Your literature review in the introduction is very limited. This would be fine if the journal was very narrowly focused, but GMD has a very broad appeal. Statisticians from the UQ (Uncertainty Quantification) community have, for example, done a lot research on surrogate modelling methods for very expensive models like an ESM. Names that come to mind are Nathan Urban (Los Alamos), Jonty Rougier (Univ. of Bristol), Michael Goldstein (Durham univ.). A series of workshops were held in Cambridge earlier this year (http://www.newton.ac.uk/event/unq/workshops) which will help you find statisticians working in this field. Non-statisticians like David Sexton at the Met office in the UK are also working on quantifying uncertainty of ESMs. There are probably lots of other research that you can also mention, but above is just a start.

When preparing the final draft of our manuscript, we will broaden the discussion of related literature.

3.       Please use more standard mathematical notation for defining vectors and matrices. In particular, please don't use $|x\rangle$ and $\langle x|$ for column and row vectors. I have never seen this notation being used before. It is much more standard to use x for a column vector and xT (i.e. the transpose of x) for a row vector.

In the final draft we will replace the Dirac notation currently used in the manuscript with $\vec{x}$ and $\vec{x}^T$.

**[GMDD](https://www.geosci-model-dev-discuss.net/)**
4.      Please also stick to normal conventions for matrix and vector algebra, e.g. page 2 line 28 you state "Occasionally we will add a matrix and a vector e.g. $B = A + |x\rangle$. Please do not do this! I know that you explain what you mean when you do this, but mathematically it is not the correct way of doing things because you're effectively defining $|x\rangle$ to be a vector sometimes and a matrix and other times. If you want to add a matrix with a vector in the way you describe then define a new matrix which has as its columns or rows the vector you want. This is a much clearer way of defining things.

In all cases $\vec{x}$ (formerly $|x\rangle$) is a vector. Writing $\mathbf{B} = \mathbf{A} + \vec{x}$ is merely a short-hand way of saying $\mathbf{B} = \mathbf{A} + \mathbf{M}(\vec{x})$, where $\mathbf{M}(\vec{x})$ is the outer product of $\vec{x}$ with a suitably-sized vector of ones (i.e., $\mathbf{M}(\vec{x}) = \vec{1}\vec{x}^T$). Making this operation explicit doesn't make the discussion any clearer; quite the contrary, it forces the reader to stop and figure out what the new operator actually does. Likewise, defining a new matrix each time we need to perform this operation proliferates symbols unnecessarily and obscures the relationship between the vector and matrix versions of the same quantity. For these reasons, we have elected to keep the broadcast notation.

5.      Page 3, line 8. You state "We refer to the ESM data as "observations" . . . ". NO, NO, NO! Please do not do this. I spent about an hour reading your paper thinking you were using real observations and then I realized I had not properly read this very important line in your methods. If you want to use your ESM output as measurements then call them "synthetic measurements". Everyone knows what this is. But if you just state measurements or observations, we think you're using the real thing.

In the final draft we will refer to the ESM output as "synthetic measurements".

[Figure]

6.    There are lots of subject specific words / jargon used here which you
assume that the reader knows the meaning of because you don't de-
fine them. Examples include: 'linear pattern scaling', 'discrete Fourier
Transform', 'randomizing the phases of G', 'spatial coherence', etc . . .

In the final draft we will include explanations of any terminology that might be
unfamiliar to a broader scientific audience.

7.    For things that you define you don't give enough detail. For exam-
ple, with EOFs (statisticians use principal components but it means the
same thing) you need to say that we normally only choose the first n
EOFs where n is determined such that most of the variance (or power
as you mentioned) is explained. Only when we get to the results sec-
tion do you talk about the number of EOFs that you're using.

We thank the reviewer for pointing out this important omission. Although principal
components are often used for dimensionality reduction, in this case we are using
them solely to diagonalize the covariance matrix. The final draft will include an
explanation of this difference.

8.    I was also completely lost with section 2.4. You seem This seems
to be key part of the methods, so really needs to be explained better.
You make statements that make no sense to the non-specialist – e.g.
in the lines prior to equation 7 you state that you're making some minor
modifications to the standard procedure, but why? And what is a zeroth
basis vector and why are you defining it in this way. These things I read
and go "okay", but I have no idea why you're doing this. This is just one
example of many scattered throughout the manuscript.

The purpose of including it is explained in detail in the paragraph following equa-
tion 7. ("This property guarantees that . . . . This property is useful because . . . .")
We will move this paragraph up before equation 7 so that the differences between

this basis vector and the others and the motivation for including it at all are clear before we begin describing how it is calculated.

9.       There's no motivation or justification for doing what you're doing. At the end of the introduction, you state that there's a need to quantify the uncertainty in ESM output. Fine. But then you jump straight into your approach of quantifying this uncertainty by generating it based on an error covariance matrix that is derived from defining the temporal and spatial correlation of the ESM output. Why is this a good way of defining the ESM uncertainty?

Our paper is not about quantifying uncertainty in ESM output at all. Instead, it is about providing a source of data, beyond what is available in public archives, for models that are *consumers* of ESM output. Uncertainty studies (in these models, not in the ESMs themselves) are just one reason why we might want to generate these datasets (we give two others in the introduction). In the final draft the opening paragraphs of the introduction will be rewritten to clarify these points.

10.       Following on from the previous point, this seems to be a major flaw in this paper. Normally when we do uncertainty analysis, we propagate the uncertainty from the inputs (e.g. uncertainty in model parameters) through to the uncertainty in the ESM outputs. Instead, you seem to be using the spatial and temporal correlations of the ESM output as a means to generate the uncertainty in the ESM output with the meta-model (or "emulator" as you call it) as the vehicle for carrying out the extra ESM runs. Perhaps I'm mistaken and this isn't what you're doing. If I am mistaken then the fact that I have misunderstood this is a major problem. If you want people to be interested in your research, you first need to communicate it clearly and (sometimes) simply to them.

As we noted above, the purpose of this paper is not to study uncertainty in ESM

outputs. It is important to realize here that the variability in ESM outputs is an important contributor to the uncertainty in downstream models *and would still be so even if there were no parameter uncertainty at all in ESMs*. Indeed, for many downstream models the uncertainty contribution from variability is *more* important than that from parameter uncertainty. As we discuss in the introduction, many ESM emulators do not produce this variability, or produce it with significant limitations. The purpose of this work is to produce an emulator that produces the full range of variability seen in the ESM data and does so in a realistic way.

The revised manuscript will explain these matters in more detail than the current manuscript.

11.     I am confused with how you train your metamodel (or "emulator" as you call it). When I train metamodel, this often involves multiple runs of the computationally expensive model (i.e. the ESM in your case). In your paper, it seems that you just need one run of the ESM to train the metamodel. Is this correct? Again, this isn't how we normally train metamodels so you need to be really clear about this. In fact there should really be a whole section in the methods explaining everything about how you constructed the metamodel. Maybe you feel what you've written is enough, but I'm just generally confused so you need to lay things out much more logically and clearly at the very least.

In section 2.2 (p.3, l. 5–7) we wrote:

> We used surface temperature data from all available 21st century runs for all four Representative Concentration Pathway (RCP) emissions scenarios (RCP2.6, RCP4.5, RCP6.0, and RCP8.5), for a total of 9 runs, each 95 years in length.

12.     Table 1 makes little sense to me. If I really concentrate I can probably understand what's going on, but you could help the reader by using

less complicated words or phrasing it in a simpler way.

Table 1 is a summary of the steps in the algorithm described in the rest of the section. Although the reviewer appears not to have found it useful, other people we have shown it to have said that the summary helped them visualize how the pieces of the algorithm fit together. Therefore, we are inclined to keep it. In the final draft of the manuscript we will expand the table caption to clarify that this is a summary of the material in the rest of the section.

13.    The results section seems to be too short (less than one page). Most papers I read have at least 3 or 4 subsections within the results section. These subsections have their own titles and help navigate the reader through the different aspects of the results. At the moment, the results just seems like a list of things. Your results show [sic] flow more like a story. You also don't really give a lot of detail, e.g. just one sentence for figure 5? What's the point of having it in there?

In dividing the paper into sections, we construed "Results" narrowly to mean "the artifacts produced by running the model". Conversely, we categorized analysis of the properties of the model output as "Discussion", which is why that section comprises 10 of the paper's 17 pages, with three subsections and two subsubsections. In the final draft we will merge the two sections into a single "Results and Discussion" section.

In addition, the final draft will describe some of the salient features in the map figures.

14.    I didn't really read through the discussion in depth, but in section 4.1.1 (and may other subsections) you describe further results that were carried out. Thse should be in the results section. The purpose of the discussion section is: (a) to give an explanation for why your results look the way they do; (b) to put your results in context of other

comparable studies. I see a bit of (a) in the discussion but no evidence of (b).

As explained above, we appear to have a different convention than the reviewer regarding the distinction between "results" and "discussion". In the final draft the two will be merged into a single section.

**2  Minor comments**

- Panels of figures: label them with letters. E.g. Figure 1a would refer to panel at the top left of figure 1.

These will be added in the final draft.

- Figures: captions lack enough details

The captions will be expanded in the final draft.

- When submitting for review, it's more helpful to put all the figures and tables at the end of the manuscript. This makes it easier for the reader to refer to a particular figure when reading a particular part of the results.

Opinions differ on this; many scientists prefer in-text figures and tables. The GMD author guidelines leave it up to the author.

---

## Author Comment (AC3) · 7 Sep 2018

**1   General comments**

- However, the language used is quite mathematical for a GMD paper. I think this could be addressed without loss of quality or conciseness.

When preparing the final draft of the manuscript, we will look for opportunities to reduce the density of the mathematics in the text. We do note, however, that it was important to us to provide enough detail for readers to be able to both recreate and evaluate the algorithm for themselves, if desired, and doing so requires a certain amount of mathematical specificity.

[Figure]

- Also, as suggested by the first reviewer, this is not an emulator in the strict sense.

It would seem that there is some diversity in the way this terminology is used in different scientific communities. Amongst the researchers who develop these kinds of models the term "emulator" seems to be preferred; therefore, we have elected to keep to that convention.

- I also agree with the first reviewer in that, ESM outputs are not "observations". "ESM outputs" would suffice.

We have adopted the first reviewer's suggestion of "synthetic measurements" to refer to the data being used to train the model.

- A related point is that the model simulates global mean surface temperature from GCMs (general circulation models/global climate models - choose your favourite acronym) rather than ESMs. The CMIP5 definition of an ESM includes an interactive carbon cycle, going from emissions to concentrations to forcing to temperature. GCMs skip the emissions step, running from prescribed concentrations that have been calculated from a simple model, e.g. MAGICC, as they were in CMIP5.

There is nothing in the model that is specific to GCMs as contrasted with ESMs. The particular input data we chose to use as a demonstration were forced by concentration, but we could equally well have selected archival datasets that were produced with the carbon cycle turned on. Since the developers of CESM refer to their model as an "earth system model", we chose to do the same, even when working with scenarios run in a mode more characteristic of a GCM.

**2 Specific comments**

- In the introduction, the application of the model to extreme events is given as a justification for its creation. However, the model only produces annual mean temperature output in each grid cell. I am not aware of an extreme indicator that uses annual mean temperatures. Such indicators are usually calculated from daily climate model output (see Zhang et al 2011, 10.1002/wcc.147). This would be a natural extension to this model, but in its current form it is not capable of analysing "extremes" in the usual sense.

This is a good point. By "extreme events" we had in mind the tails of the distribution of annually averaged values. We will adjust the language in the final draft to clarify what we had in mind.

- I don't disagree with the authors about the notation convention: I understand the broadcasting concept used in their convention and agree it aids readability. I do find it hard to follow the equations though. If we have $|T_g\rangle = \mathbf{O}|\lambda\rangle$, then this suggests to me that $|T_g\rangle$ is a column vector of shape 855 x 1 formed by multiplication of $\mathbf{O}$ (855 x 55296) by $|\lambda\rangle$ (55296 x 1). In eq(2) you have $T_g|w\rangle + |b\rangle$. Is $T_g$ (not bracketed in eq(2)) times $|w\rangle$ a column vector times a column vector? ? How is this defined?

$T_g$ (without brackets) is a scalar. On the other hand, $|T_g\rangle$ is a vector of global mean temperature values. When defined by the first equation in the quote, this vector is made up of the values of $T_g$ for each year of each model in the input set. In other words, the name of a variable tells us what physical quantity the variable represents, and the decoration tells us how many we have and what kind of structure they are organized into. We will add some clarifying remarks on this point to the notation section.

- And then in equation 3, there is $|T_g\rangle$ (a column) times $\langle w|$ (a row), which I think is 855 x 855, then added to $|b\rangle$ (855 x 1)? and subtracted from $\mathbf{O}$ (55296 x 855 - but how is this broadcasted?) If there are no typos in these equations, it would be helpful here to put in a diagram of the matrix dimensions in the equations 1 to 3.

The symbol $\langle w|$ is a row vector, with dimension 1 x 55296 (i.e., one value for each grid cell). The product $|T_g\rangle\langle w|$ is an outer product, the result of which is a matrix $(855 \times 1) \cdot (1 \times 55296) = (855 \times 55296)$. The vector $|b\rangle$ likewise has dimension (55296 x 1) (again, one value for each grid cell). Because this matches the number of columns in the matrix formed by the outer product, it can be broadcast in the usual way. The result is still a matrix (855 x 55296), which is confomant with the matrix $\mathbf{O}$ that it is being subtracted from.

We will clarify the dimensions of the vectors of pattern scaling coefficients, and we will add a figure that shows how these quantities fit together to produce the final matrix of residuals.

- $\sigma$ values in table 1 and p5 line 9. I think these are the singular values of $\mathbf{R}$, but it is not really explained what these are or what they mean. This paragraph could do with some expansion of the key terms (rank deficient, discrete Fourier transform). Does dropping EOFs where $\sigma < \sigma_{threshold}$ guarantee full rank?

The $\sigma$ are the singular values; we will clarify this in the final draft. We will also provide a brief explanation of what the singular values mean, and we will supply a reference to an approachable introduction to Fourier transforms and their applications.

Technically, having all $\sigma > 0$ is enough to guarantee full rank, so it would be more correct to say that the problem here is ill-conditioning, rather than rank deficiency. However, because we do not use the SVD to invert the matrix (only to find the

principal components), it is not clear that the ill-conditioning causes any particular harm. Therefore, in the final draft we will regard the dropping of components with very small singular values as an implementation detail and omit the discussion of it in the text.

- Section 3: Can the four images in figure 1 be interpreted as ensemble members? If so, it would be good to state this.

Yes, they can. We will comment on this in the final draft.

- figures 4-6 and associated discussion in lines 24-28 on page 6: The periodic variability in EOFs 2, 3 and 5 - could these have a physical interpretation? For example there seems to be an El Nino style feature in EOFs 3 and 5. On the other hand, is there any evidence that the lower EOFs are not just noise?

We, too, had noticed the resemblance to El Nino in those components; however, it wasn't clear how to make a rigorous comparison between the patterns we see here and real-world El Nino events (since surface air temperature isn't really the right variable for computing a proper El Nino index). Developing a methodology for making such a comparison is outside the scope of this paper (though it would be interesting research in its own right), so we decided to characterize these components generically as periodic modes of variability, rather than to attribute a physical cause to them.

We feel very confident that the lower EOFs (we assume that by this you mean the ones with lower total power, not the ones earlier in the sequence) *are* mostly noise. In the time dimension their power spectra are almost completely flat, and the length scale of spatial correlations is just a few pixels. This is pretty much the definition of "noise" in this context. That said, there is still *some* structure, even in these noisy basis functions, and characterizing that structure with this

model allows us to ensure that the noise in the output realizations has the *same* structure.

To put it another way, you could probably get an adequate representation of the noise in the system by just applying a random perturbation (i.e., without regard to space or time correlation) and then running a smoothing kernel over the result so as to reproduce the short-range correlations observed in the noisy components. But, what should be the width of that kernel, and how should that noise field be weighted relative to the structured components? Those things are an important part what we are trying to model with this technique.

- Section 4.2 got me thinking that as the model is trained on the RCP outputs, is there any difference in the results when taking just the set of realisations from RCP2.6 and RCP8.5? Certaintly across ESMs, the variance across models increases with increasing global mean temperature. It would therefore not be correct to use a variability model that is trained on RCP8.5 for low forcing scenarios or those with a peak and decline. I note the authors address this in section 4.3, but I wonder if they have tested this.

We have worked with models trained on a single scenario, and for the most part the results are qualitatively similar to the multi-scenario results. We didn't try to run any statistical tests to detect differences, but with the limited amount of data available it seems unlikely that any such differences would be detectable. Therefore, although it's theoretically possible that by using variability from a model trained solely on a scenario of interest (supposing you know in advance what scenario that is) you might get more accurate results for that scenario. However, in practice the difference is likely to be small and perhaps offset by the effects of having less data to train on. Many of these topics would be worth revisiting in the future, particularly once improvements in the mean field response are in place.

**3 Technical Corrections**

- page 5, line 3: allow → allows

  Thanks. We will correct this.

- page 6, line 3: 143 seconds. What is the machine architecture here?

  This was on a midrange workstation. We will mention this in the final draft.

---

## Author Response (AR2)

**Fldgen v1.0:  An Emulator with Internal Variability and Space-Time Correlation for Earth System Models**

Robert Link[1], Abigail Snyder[1], Cary Lynch[2], Corinne Hartin[1], Ben Kravitz[3,4], and Ben Bond-Lamberty[1]

[1]Pacific Northwest National Laboratory, Joint Global Change Research Institute, 5825 University Research Ct., College Park, MD, USA

[2]Connecticut Department of Energy and Environmental Protection, 10 Franklin Square New Britain, CT, USA

[3]Department of Earth and Atmospheric Sciences, Indiana University, 1001 E. 10th St., Bloomington, IN, USA

[4]Atmospheric Sciences and Global Change Division, Pacific Northwest National Laboratory, 902 Battelle Boulevard, Richland, WA, USA

**Correspondence:** Robert Link (robert.link@pnnl.gov)

**Authors' responses to reviewers' comments**

**Anonymous Reviewer #2**

- The size of the vector b in figure 1 should be (55296 x 1), shouldn't it? The reviewers' responses suggest so, and working through the calculation and text myself, this is how I understand it.

5      An aside: I know the software is written in R, but the broadcasting explaination in section 2.1 invokes numpy. Technically, numpy broadcasting would only work here if the array b was of size (55296). The text implies that the array size is (55296 x 1), which is fundamentally different in numpy. As a Python native this confused me for a while, but the authors have explained their notation and broadcasting choice in section 2.1, so I am not too concerned about this.

10    Both of these observations are correct. I have fixed Figure 1 accordingly, including dropping the second index on $b$. However, $b$ is still depicted graphically as a column vector, since that seems to be how most people regard "vectors" of unspecified shape.

- page 7, line 17: 3 to 20 years. I think 3 to 5 years is a bit more apparent; I don't see any noticeable periodicity on frequencies < 0.2. The captions to figures 5 and 6 also say 3-5 years.

The skewed shape of the PSD for EOF-2 makes it a little nebulous what the lower bounds of the frequencies in that mode are.

15    I have changed the frequency range in the description to 3–5 years to coincide more closely with the peak of the PSD and for consistency with the figure captions.

    Thank you for your helpful suggestions.

**Anonymous Reviewer #3**

The title is not very informative - all emulators are computationally efficient, so what marks this one out? A title that refers to internal variability might be useful, and would help to draw the attention of target end-users?

Specific examples of cases when impacts are strongly dependent upon internal variability might be useful to make it crystal clear what this is doing and why it is needed. Even just pulling in the headline result from Ray "Climate variation explains a third of global crop yield variability" would help non-specialist readers a lot I think.

These are both excellent suggestions, which I have implemented. Thank you.

I had similar thoughts as reviewer two re using different RCPs used to train the emulator, and I didn't find the response fully convincing – some tests should be possible here? In particular, I found myself wondering whether EOF1 reflected responses to different RCPs. Did you check whether EOF1 was similar in an emulator trained on a few simulations with a single RCP (and didn't disappear, as when you trained on a single simulation)? Why not compare the variance of the full emulator (i.e. trained on all 9 simulations) separately with the RCP2.6 simulations and with the RCP8.5 simulations to quantify any scenario bias? In any event, some discussion of this should be included in the text as I suspect many people will question it. Repeating from Reviewer Two "Section 4.2 got me thinking that as the model is trained on the RCP outputs, is there any difference in the results when taking just the set of realisations from RCP2.6 and RCP8.5? Certaintly across ESMs, the variance across models increases with increasing global mean temperature. It would therefore not be correct to use a variability model that is trained on RCP8.5 for low forcing scenarios or those with a peak and decline. I note the authors address this in section 4.3, but I wonder if they have tested this."

My analysis of this point wound up being rather long, so I have placed it in our public code and data archive. It can be downloaded from https://zenodo.org/record/2586040/files/cc-analysis.nb.html?download=1 The summary is that the mean response model trained on the ensemble members from a single RCP is practically indistinguishable from the mean response model trained on an equivalent number of ensemble members from different RCPs. I have added a new subsction 3.4, which compares these mean field models and directs readers to the archive for further tests.

[revised manuscript text omitted]